# Neurotrophic Effect of Fish-Lecithin Based Nanoliposomes on Cortical Neurons

**DOI:** 10.3390/md17070406

**Published:** 2019-07-09

**Authors:** Catherine Malaplate, Aurelia Poerio, Marion Huguet, Claire Soligot, Elodie Passeri, Cyril J. F. Kahn, Michel Linder, Elmira Arab-Tehrany, Frances T. Yen

**Affiliations:** 1Research Unit Animal and Functionality of Animal Products, Quality of Diet and Aging Team (UR AFPA) Laboratory, Qualivie team, University of Lorraine, 54505 Vandoeuvre-lès-Nancy, France; 2LIBio Laboratory, University of Lorraine, 54505 Vandoeuvre-lès-Nancy, France

**Keywords:** n-3 fatty acids, nanoparticles, brain

## Abstract

Lipids play multiple roles in preserving neuronal function and synaptic plasticity, and polyunsaturated fatty acids (PUFAs) have been of particular interest in optimizing synaptic membrane organization and function. We developed a green-based methodology to prepare nanoliposomes (NL) from lecithin that was extracted from fish head by-products. These NL range between 100–120 nm in diameter, with an n-3/n-6 fatty acid ratio of 8.88. The high content of n-3 PUFA (46.3% of total fatty acid content) and docosahexanoic acid (26%) in these NL represented a means for enrichment of neuronal membranes that are potentially beneficial for neuronal growth and synaptogenesis. To test this, the primary cultures of rat embryo cortical neurons were incubated with NL on day 3 post-culture for 24 h, followed by immunoblots or immunofluorescence to evaluate the NL effects on synaptogenesis, axonal growth, and dendrite formation. The results revealed that NL-treated cells displayed a level of neurite outgrowth and arborization on day 4 that was similar to those of untreated cells on day 5 and 6, suggesting accelerated synapse formation and neuronal development in the presence of NL. We propose that fish-derived NL, by virtue of their n-3 PUFA profile and neurotrophic effects, represent a new innovative bioactive vector for developing preventive or curative treatments for neurodegenerative diseases.

## 1. Introduction

Polyunsaturated fatty acids (PUFA) have been extensively studied for their effects in neuronal development and growth, where investigations have particularly focused on n-3 PUFA for their therapeutic effects on neurodegenerative diseases, such as Alzheimer’s disease. Both n-3 and n-6 PUFA have significant roles in membrane structure and function and in cell signaling. They also play important roles as substrates for the synthesis of lipid mediators that are involved in inflammation. Indeed, the consumption of n-3 PUFA docosahexanoic (DHA) and eicosapentaenoic (EPA) acids has been shown to decrease the amounts of n-6 arachidonic acid (AA), which is a precursor of the production of eicosanoids that are involved in inflammation [1]. DHA and EPA are precursors of the anti-inflammatory resolvins, and they may also exert an anti-oxidative role by modulating the activity of proteins that are involved in oxidative stress in the central nervous system [2]. PUFAs are also known to be directly involved in the regulation of gene expression in the brain [3].

Numerous studies have focused particularly on n-3 PUFA’s role on membrane structure and function. Indeed, DHA represents 60% of PUFA in neuronal membranes. Lipid rafts are needed for axon guidance, allowing for cone growth and cell polarization of neurons for the formation of new synaptic connections [4]. The growth-associated protein 43 (GAP43) is an abundant raft-protein that is involved in the development of axons and nerve growth, and its expression can be induced by fatty acids, such as DHA and oleic acid, in immunofluorescence studies [5,6]. Both cell and animal studies have also demonstrated increased neuronal differentiation and neurite outgrowth in the presence of PUFA [7,8].

It is clear that n-3 PUFA provide beneficial effects in neuronal development and growth by increasing membrane fluidity [9,10,11], or as activators of cell signaling functions [3,12]. Furthermore, by improving synaptic plasticity and the formation of new synaptic connections, they are key in the prevention of neuronal damage that can be associated with aging or neurodegenerative diseases, such as Alzheimer’s disease. 

These fatty acids must be obtained either directly from the diet, or are produced by synthesis in the liver from precursors linoleic and α-linoleic acid before transport and delivery to the brain, since neurons lack the enzymes for the *de novo* synthesis of DHA and arachidonic acid. Providing the means for the efficient delivery of PUFA to the brain would thus be of interest towards maintaining brain membrane integrity and synaptic function in the aging process for preventive purposes and also as adjuvant for better response to therapeutics [13]. 

We have developed a green extraction technique to prepare nanoliposomes from natural resources (patent n° FR 2.835.703, [14]). These soft nanoparticles contain multiple bilayers that are composed of natural lecithin surrounding an aqueous compartment. Its unique physicochemical characteristics allows the incorporation of hydrophilic, amphiphilic, or hydrophobic molecules, and, by virtue of the lipid bilayer, these nanoliposomes can fuse with cell membranes, thereby delivering the liposome contents to the target tissue. Nanoliposomes the offer advantages of low toxicity, of being modifiable with respect to size and surface, biocompatibility, and biodegradability [15]. Nanoliposomes that are prepared using lecithin derived from fish by-products, which represent a major source of n-3 PUFA, show positive effects on cell proliferation with a stimulation of cell activity in mesenchymal stem cells [16], as well as in neuronal cells [17]. Here, we report the further characterization of these nanoliposomes. Increased metabolic mitochondrial activity and neural network formation was observed in the presence of nanoliposomes [17], suggesting the potential effects on synapse formation and neurogenesis. It remained to be determined whether the nanoliposome-induced effects were neuron-specific, since the cells used were shown to express the glial fibrillary acidic protein GFAP which is found only in glial cells. Here, we investigate this further by examining the effect of nanoliposomes on specific synaptic and neurogenesis markers in purified primary culture preparations of rat embryo cortical neurons.

## 2. Results and Discussion

### 2.1. Lipid Classes

In salmon lecithin, phosphatidylcholine was the major class of phospholipids and it represented 42% of this fraction (Table 1). The triacylglycerol (TAG) content in lecithin was 31.20 ± 0.4% and the amount of polar fraction was 67.65 ± 0.9%. A small amount of cholesterol was detected in the fraction, at a level of 1.15 ± 0.1%.

### 2.2. Fatty Acid Analyses

Analyses by gas chromatography revealed that fatty acid composition of salmon lecithin consists of a variety of polyunsaturated fatty acids (PUFA) (Table 2). A number of PUFAs of omega 3 and omega 6 were identified in salmon lecithin, of which C22:6 n-3 (26.26%) and C20:5 n-3 (11.03%) were the most significant. C18:1 n-9 (13.87%) and C16:0 (20.67%) were the two major monounsaturated and saturated fatty acids, respectively. The n-3/n-6 ratio was 8.88 with a ratio of DHA/EPA of 2.38.

### 2.3. Liposome Size and Electrophoretic Mobility Measurements and Morphological Properties

The particle sizes of different nanoliposome preparations were immediately measured after sonication. The liposome size depends on the viscosity of the material and agitation parameters, including sonication amplitude and time. The size of the nanoliposome also depends on fatty acid composition (the presence of LC-PUFA), lipid profile, and the surface-active properties of the lecithin, like as the percentage and type of polar head [16,18]. The hydrodynamic diameter of nanoliposomes for salmon lecithin was 110 ± 0.32 nm, and the polydispersity index was 0.29 ± 0.00. The polar fraction of lecithin (phosphate residues) has a negative charge. The electrophoretic mobility of nanoliposomes from salmon lecithin was −3.5 ± 0.09 μmcm/Vs with a relatively high stability, According to the electrophoretic light scattering (ELS) results, according to the electrophoretic light scattering (ELS) results. Different types of phospholipids, such as phosphatidylserine (PS), phosphatidic acid (PA), phosphatidylglycerol (PG), phosphatidylinositol (PI), phosphatidylethanolamine (PE), and phosphatidylcholine (PC) were present in salmon lecithin. Their charge is negative, except for PC and PE, which does not exhibit a net charge at physiological pH. Therefore, these anionic fractions most likely account for the negative electrophoretic mobility [19]. Nanoliposome stability measured by particle size at 4 °C and 37 °C over a one-month period showed no significant variation between day 0 and day 30. 

The images that were obtained by Transmission Electron Microscopy (TEM) showed that prepared nanoliposomes were in the form of multilamellar vesicles (MLV). The bilayer nature of the vesicles was visible in these micrographs (Figure 1). We also observed some droplets in each formulation, because of the presence of small quantities of oil (10%). We used Atomic Force Microscopy (AFM) in order to observe the morphological property in nanometric scale. The results showed that, despite the variety of fatty acids in salmon lecithin, we did not observe phase segregation.

### 2.4. Membrane Fluidity

Membrane fluidity of nanoliposome is one of the important parameters that affect the release of active molecules and drugs from nanoliposomes, which reflects the order and dynamics of phospholipid alkyl chains [20]. Membrane fluidity depends on the lipid composition of nanoliposomes. By increasing the percentage of unsaturated FAs, the packing between the phospholipids decreases and keeps the level of hydration, thus maintaining membrane fluidity [21].

Nanoliposomes that were prepared from salmon lecithin had a membrane fluidity of 3.20 ± 0.03 when they contained higher proportions of PUFA. The lipid fluidity was expected to increase the permeant diffusion rate and partitioning tendency, which yields a possible squaring effect on the enhancement factor.

### 2.5. Treatment of Primary Cultured Neurons with Nanoliposomes

We investigated the effect of these lipid particles on the maturation of primary cultures of rat cortical neurons in view of the high amounts of n-3 fatty acids. Neurons were prepared and plated on day 0 in vitro (D0). To assess potential cytotoxic effects of NL, a set of cultured cells were treated with increasing concentrations of NL on D3, and incubated 24 h. MTT assay on D4 revealed a small decrease in cell viability at 50 µg/mL NL, which became statistically significant at NL concentrations > 100 µg/mL (Figure 2). Cell viability was not significantly changed in the presence of 10 µg/mL NL, which was therefore selected as the concentration to be used for the study. 

The primary cultures of neurons were prepared and then incubated in the absence or presence of NL. Since significant changes in the maturation process of the neuronal network are observed after D3, cell culture media was replaced with the same media supplemented with 10 µg/mL NL on D3. The cell lysates were then prepared on D4 for immunoblotting studies to assess the changes in protein levels of different markers of neuronal functions. A similar set of cells that were treated in an identical manner, but cultured in the absence of NL, were used as controls that allowed us to establish the baseline; cell lysates were prepared on D4, D5, and D6. The results in Figure 3 revealed that there was a general tendency of higher protein levels of markers of neuronal and cone growth (β-tubulin and GAP43) in control cells without NL, as well as synaptic proteins (SNAP25, PSD95, synaptophysin), when comparing the protein levels on lysates from D5 or D6 to those of D4, with the exception of the post-synaptic PSD95 marker on D6. The two structural markers, β-tubulin and GAP43 protein levels, also showed similar profiles when compared with D5 or D6 to D4. The latter protein is indicative of cytoskeletal development that is conducive with axonal and dendritic growth. Interestingly, in cells that were incubated in the presence of NL on D3, the protein profiles appeared to be similar to that of D5 or D6, rather than D4 control cells in the absence of NL, with the exception of synaptophysin, which suggests the accelerated maturation of these cells in the presence of the NL. Unfortunately, data analysis did not reveal any statistically significant differences, with the exception of synaptophysin levels between D4 and D5. This could be due, in part, to the variation of protein levels between the different culture preparations, despite a similar trend in three separate cultures, as shown in the representative immunoblot (Figure 3A). Immunofluorescence studies were performed on cells that were labeled with the different markers to evaluate the morphology of the neurons during culture since the maturation of a neuron involves several stages including growth cone, axon, and synaptic button formation.

Image analysis (Figure 4A) showed that GAP43 expression increased with neuronal extensions and could be seen as being granular with spots in control cells on D5 as compared to those on D4. This protein is transported along neurons from the cellular body to cone growth and therefore is a maker for neuronal growth. On D6, GAP43 was present primarily as granular fluorescent spots that were localized at growth cones and axons, most likely participating in the formation of new extensions. Cells that were incubated with NL exhibited a GAP43 profile on D4 that was similar to that of control cells on D5, with staining in both neuronal extensions and as granular with spots, indicating the accelerated state of outgrowth in the presence of NL as compared to untreated cells.

With neuronal growth, the formation of synapses should be observed. Co-labeling of the pre- and post-synaptic markers PSD95 (green) and SNAP25 (red) would normally indicate the formation of a functional synapse with both pre- and post-synaptic button formations, which immunoblots could not detect. Image analyses to count spots on ImageJ revealed a higher number of SNAP25 (granular red puncta) as compared to that of PSD95 (granular green puncta) in untreated cells on D4 (Figure 4B). The merging of images revealed as orange spots were indicative of colocalization of both markers, and they were also detected in untreated cells on D4 and increased when compared to values that were obtained for D5 and D6 combined. In cells that were treated with NL, statistically significant increases in SNAP25 and PSD95 individually, as well as SNAP25 + PSD95 orange spots were observed, as compared to untreated cells (D4), which is indicative of a higher number of synapse formations in the presence of these lipid particles.

The presence of syntaxin, as part of the SNARE complex located at cell membrane synaptic terminals, was detected in neurons, appearing to be transported along the neuronal extensions. Synaptophysin, which is part of a complex with synaptobrevin and is important in neuronal development, is localized around the cellular body and along the axons on D4. While syntaxin labeling was already granular on D5, that of synaptophysin remained detectable along the axons, which may be due to the different roles and functional localizations. On D6, syntaxin labeling was found around the cellular body, which could indicate the continuous synthesis of this protein. Labeling of dynamin I, a GTPase that is involved in synaptic vesicle recycling, was very similar to that of synaptophysin. In NL-treated cells, syntaxin, synaptophysin and dynamin labeling profiles were very similar to those of D5 and D6, detected on both axons and around the cellular bodies. 

Embryonic cortical neuron development in vitro proceeds through several stages that are identified by morphological changes that occur during maturation. Neurons establish distinct compartments that initially form as neuritis as they grow and differentiate, which become axons, and finally lead to the formation of synaptic contacts through dendritic spines and axon terminals to establish the neural network [22]. Image analysis of labeled cells by immunofluorescence revealed the different structural and morphological changes that occur between D4 and D6 of culture consistent with neuronal development and differentiation, including neurite outgrowth and synaptogenesis, as suggested by the tendencies that were observed in the immunoblots. Preliminary data indicate that the presence of NL may also activate cell survival pathways (Poerio et al, unpublished data), which suggests the potential neuroprotective properties of these NL. In summary, these results indicate that the presence of NL accelerates the maturation processes of embryonic rat cortical neurons in primary culture. 

## 3. Materials and Methods

Salmon lecithin was obtained by enzymatic hydrolysis in our laboratory, as described before [23]. Lipidic fractions were extracted by a low temperature enzymatic process that is solvent-free. The following materials were used in the present study: acetonitrile and diethyl ether (Sigma-Aldrich, Lyon, France), boron trifluoride-methanol (BF3) (Supelco, Bellfonte, PA, USA), chloroform (VWR-Prolabo, Milan, Italy), hexane, methanol, and formic acid (Carlo-Erba, Peypin, France), and ammoniac (Merck KGaA, Darmstadt, Germany). All of the organic solvents used were analytical grade reagents.

### 3.1. Fatty Acids Composition

Fatty acid methyl esters (FAMEs) from salmon lecithin were prepared as described by Ackman (1998) [24]. Subsequently, FAMEs were analyzed while using a Shimadzu 2010 gas chromatography (Shimadzu, Marne-la-Vallée, France) system that was equipped with a flame-ionization detector. FAME was injected into a fused silica capillary column (60 m, 0.25 mm i.d. × 0.20 µm film thicknesses, SPTM2380 Supelco, Bellfonte, PA, USA). Injector and detector temperatures were settled at 250 °C. The column temperature was fixed initially at 120 °C for 3 min., then raised to 180 °C at a rate of 2 °C min.^−1^, and maintained at 220 °C for 25 min. Individual fatty acids were identified while using standard mixtures (PUFA1 from a marine source and PUFA2 from a vegetable source; Supelco, Sigma-Aldrich, Bellefonte, PA, USA). The results are shown as mean ± SD of triplicate analyses.

### 3.2. Lipid Classes

The lipid classes of different lipid fractions from salmon lecithin were determined while using the Iatroscan MK-5 TLC-FID (Iatron Laboratories Inc., Tokyo, Japan). The measurement was performed according to the protocol that is described in detail in our previous paper [25]. Two migrations were done to determine the proportion of neutral and polar lipid fractions. All of the standards were purchased from Sigma (Sigma– Aldrich Chemie GmbH, Munich, Germany). Area percentages of each pic are presented as the mean ± SD of three repetitions.

### 3.3. Preparation of Nanoliposomes

Salmon lecithin was extracted by a controlled enzymatic procedure, according to the Linder et al. Patent [14] that was slightly modified. The use of a protease, namely Alcalase 2.4L (Novo Nordisk, Bagsvaerd, Denmark), lead to recovering most of the lipid fraction after a controlled hydrolysis degree by the pH-stat method [26]. The protease was inactivated by heating at 90 °C during 15 min. before a centrifugation step (85 °C, 15 min. 4000× *g*). A freeze-dried step of the substrate, containing the lipid and protein fractions, was done, and acetone at −18 °C was added to separate the lecithin and neutral lipids, according to Hasan et al. [27].

One gram of lecithin was dissolved in ethanol, and then a thin lipid film was formed on the wall of the flask by means of a Rotavapor (Laborita 4000 Heidolph, UK) and by completely evaporating the organic solvent under vacuum, followed by hydration with 49 mL of distilled water, and the suspension was agitated for 5 h under nitrogen. The samples were then sonicated (sonicator probe, Sonics & Materials Inc., CT, USA) at 40 KHz and 40% of full power for 360 s (1 s on, 1 s off) to obtain a homogeneous solution. The liposome samples were stored in glass bottles in the dark at 4 °C.

### 3.4. Liposome Size and Electrophoretic Mobility Measurements

The mean size and electrophoretic mobility of liposomes were measured by dynamic light scattering (DLS) while using a Malvern Zetasizer Nano ZS (Malvern Instruments Ltd, UK). The samples were diluted (1:200) with ultrapure distilled water prior to measuring size and electrophoretic mobility. The size distribution of particle as well as the dispersed particles electrophoretic mobility was measured to evaluate the surface net charge around droplets. Measurements were made at 25 °C with a fixed scattering angle of 173°, the refractive index (RI) at 1.471, and absorbance at 0.01. The presented sizes are the z-average mean (dz) for the liposomal hydrodynamic diameter (nm). The measurements of electrophoretic mobility were performed in standard capillary electrophoresis cells that were equipped with gold electrodes at the same temperature. At least three independent measurements were performed for each condition.

### 3.5. Stability of Nanoliposomes

The nanoliposomes were stored in a drying-cupboard at 37 °C for 30 days. Mean particle size, electrophoretic mobility, and polydispersity index of all formulations were analyzed every three days. The same protocol described previously was used for each analysis.

### 3.6. Transmission Electron Microscopy (TEM)

Nanoliposomes were negatively stained according to the protocol of Colas et al. (2007) [28] and then visualized on TEM. Briefly, the samples were diluted 25-fold with distilled water to reduce the concentration of the particles. The same volume of the diluted solution was mixed with an aqueous solution of ammonium molybdate (2%) as a negative staining agent. The samples were examined using a Philips CM20 Transmission Electron Microscope (Philips, Dresden, Germany) associated with an Olympus TEM CCD camera after 3 min. incubation at room temperature and 5 min. incubation on a copper mesh coated with carbon. 

### 3.7. Atomic Force Microscopy Imaging

An Image of the supported lipid bilayers (SLBs) was acquired on a Bruker AFM Dimension FastScan (Bruker, Billerica, MA, USA) with NPG tips (Bruker, Billerica, MA, USA) with a spring constant of about 0.32 N/m (manufacturer data). An image was obtained at room temperature in Peak Force QNM mode, shortly after SLB formation. An image of 5 µm size was acquired at least for two different samples and two different areas per sample. Images were analyzed using Nanoscope Analysis (v140r2, Bruker, Billerica, MA, USA). 

### 3.8. Membrane Fluidity of Nanoliposomes

The membrane fluidity of all samples was measured by fluorescence anisotropy. TMA–DPH was used as fluorescent probe, which is a compound that contains a cationic trimethylammonium (TMA) substitute that acts as a surface anchor to improve the localization of the fluorescent probe of membrane interiors, DPH. This measurement was carried out according to the method that was described by Maherani et al. [29]. Briefly, the solution of TMA–DPH (1 mM in ethanol) was added to the liposome suspension to reach a final concentration of 4 µM and 0.2 mg/mL for the probe and the lipid, respectively. The mixture was lightly stirred for at least 1 h at room temperature and protected from light. Subsequently, 180 µL of the solution was distributed into each well of a 96-well black microplate. The fluorescent probe was vertically and horizontally oriented in the lipid bilayer. The fluorescent intensity of the samples was measured with Tescan INFINITE 200R PRO that was equipped with fluorescent polarizers. The samples were excited at 360 nm and emission was recorded at 430 nm under constant stirring at 25 °C. The Magellan 7 software was used for data analysis. The polarization value (P) of TMA–DPH was calculated while using the following equation:
(1)$$P=\frac{I_{∥}-GI_{⊥}}{I_{∥}+2GI_{⊥}},$$
where I_‖_ is the fluorescent intensity parallel to the excitation plane, I_⊥_ is the fluorescent intensity perpendicular to the excitation plane, and G is the factor that accounts for transmission efficiency. Membrane fluidity is defined as 1/P. The results were presented as mean ± SD of triplicate analyses.

### 3.9. Cell Culture

Cell culture studies were performed at the Bioavailability–Bioactivity (Bio-DA) platform. The primary cultures of cortical neurons from rat fetuses (embryonic day 16–17) were prepared, as described previously [30]. Cells were plated at 10 × 10^4^ cells/cm^2^ onto plates or slides pre-coated with poly-L-ornithine (15 µg/mL for plates, 150 µg/mL for slides, Sigma), and culture in neuronal culture medium M2 composed of DMEM-F12 (Invitrogen, Illkirch, France) medium containing 0.5 µM insulin, 60 µM putrescine, 30 nM sodium selenite, 100 µM transferrin, 10 nM progesterone, and 0.1% (*w*/*v*) ovalbumin (Sigma). The cell cultures were maintained at 35 °C in 6% CO_2_. The absence of glial fibrillary acidic protein (GFAP) following immunoblotting of cell lysates verified the purity of the neurons (data not shown). Neurons were incubated for up to five days, where day (D) 0 is the day of the preparation of the primary culture. The cells were incubated in the presence of NL on Day 3 for 24 h. Cytotoxicity of NL on cells was determined while using the MTT assay, as described previously [30]. 

### 3.10. Immunoblotting

On D4, D5, and D6, the cells were washed with ice-cold PBS (Invitrogen). Cell lysates were prepared using RIPA buffer, as previously described [30]. Equal volumes of cell lysates were applied to SDS-PAGE, and transferred to PVDF for immunoblotting [30]. The following antibodies were used at 1/1000 dilution: anti-synaptophysin (Cell Signaling 4329S), anti-SNAP25 (Santa Cruz sc-7538), anti-PSD95 (Cell Signaling, 2507S), anti-GAP43 (Cell signaling, 8945S), and anti-β-tubulin (Sigma T5201). Secondary antibodies, anti-mouse HRP-linked, and anti-rabbit HRP linked from Cell Signaling and anti-goat HRP-linked from Santa Cruz were used to detect the bands with Supersignal chemiluminescence ECL kit (Millipore, Molsheim, France). Gel images were obtained while using the Fusion Imaging System (Fusion Fx5; Vilber Lourmat, France), and densitometric analysis was performed using the freeware ImageJ. 

### 3.11. Immunofluorescence Analyses

Cortical neurons cultured on slides were fixed in PBS containing 4% (*w*/*v*) paraformaldehyde for 30 min. at room temperature, and then permeabilized by incubation with Dulbecco’s PBS (DPBS, Invitrogen) containing 30% Cas-Block and 0.2% (*v*/*v*) Triton X-100. The cells were incubated with the primary antibody diluted in the same solution for 2 h at room temperature. After washing with DPBS, the slides were incubated in the dark in the presence of secondary antibodies that were conjugated to a fluorophore in DBPS. Cell nuclei were stained using 4,6-diamidino-2-phenylindole (DAPI). Images were obtained while using a confocal microscope (Fluoview Fv10i, Olympus, Rungis, France); analysis of SNAP25 and PSD95 spots was performed using ImageJ (particle analysis after subtraction of background; spots counted were visually verified using the outline option).

### 3.12. Statistical Analyses

All of the results are shown as the mean ± SD, unless otherwise indicated. The values for each set of cells that were treated with NL were calculated as % of control cells incubated in the absence of NL from the same experiment before statistical analysis. Statistical analysis was performed while using ANOVA followed by a Scheffe’s post hoc test; significance was considered as *p* < 0.05.

## 4. Conclusions

In conclusion, these results demonstrate that the supplementation of NL in the culture media accelerate the development of neural networks in primary cultures of rat cortical neurons. Treatment with NL appeared to accelerate neuronal development, neurite outgrowth, and synaptogenesis, thus demonstrating a beneficial role of these pure NL on neuronal development by virtue of their intrinsic properties, which may also include neuroprotective effects. We would propose that, as multilayer vesicles with improved fusogenic properties, these PUFA-rich NL could represent a means towards providing cells with PUFA in a synergistic and efficient way, rather than directly using PUFA as fatty acids. Indeed, PUFA supplementation may be a necessary adjuvant for the aging brain to be more responsive to the therapeutic treatment of age-related neurodegenerative diseases, such as Alzheimer’s disease [13]. Furthermore, nanoliposomes can also serve as a carrier of active molecules, thus serving a dual role in not only in providing the necessary lipid components for optimal membrane organization, but also in the delivery of active molecules for preventive or curative treatments of neurodegenerative diseases.

## Figures and Tables

**Figure 1 marinedrugs-17-00406-f001:**
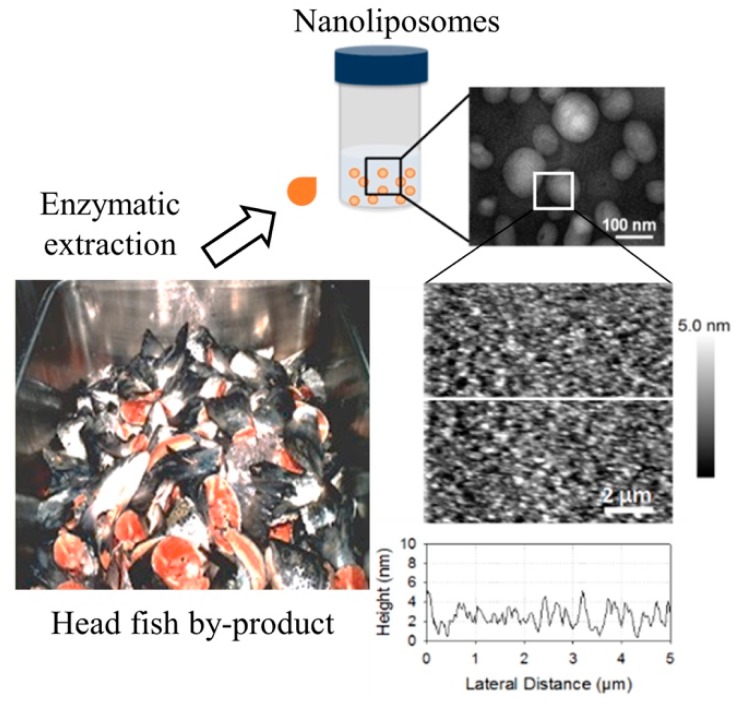
Formation and characterization of nanoliposomes from natural lecithin. Schematic diagram of the study, demonstrating the extraction of lecithin from marine source followed by generation of nanoliposomes; the inset shows a representative Transmission Electron Microscopy (TEM) image of the fabricated nanoliposomes and Atomic Force Microscopy (AFM) images of supported lipid bilayers made of phospholipids from salmon lecithin.

**Figure 2 marinedrugs-17-00406-f002:**
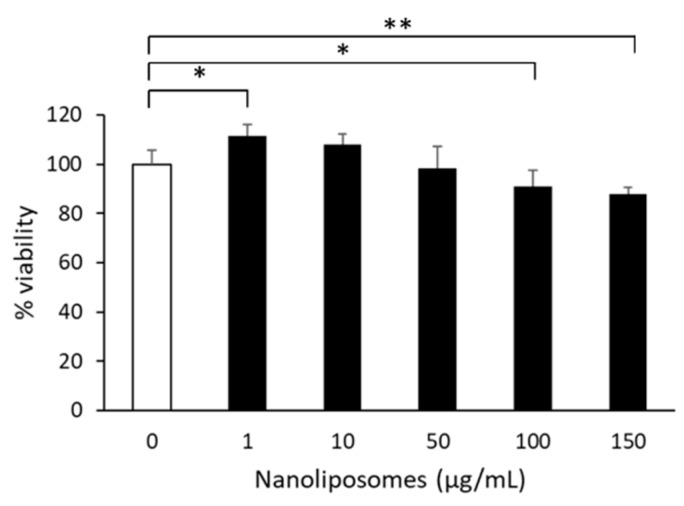
Cytotoxicity of nanoliposomes. Primary cultures of rat embryo cortical neurons were incubated in medium containing the indicated concentrations of nanoliposomes on day 3 in vitro. After 24 h, MTT assay was performed to assess cell viability. Mean and SD are shown for quadruplicate determinations. Statistical significance is shown as compared to that obtained for cells incubated in absence of nanoliposomes (* *p* < 0.05, ** *p* < 0.01).

**Figure 3 marinedrugs-17-00406-f003:**
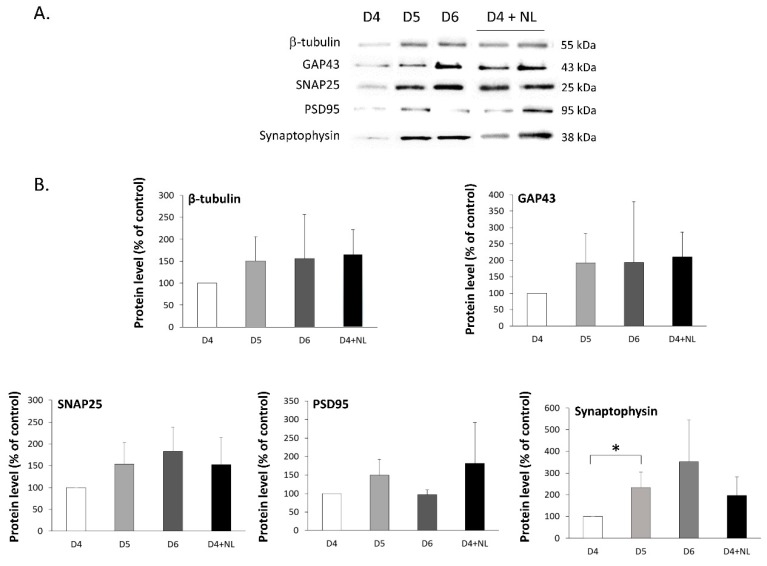
Immunoblots of pre- and post-synaptic markers in primary cultures of cortical neurons treated with fish nanoliposomes (NL). (**A**) Representative immunoblots of cell lysates prepared from primary cultures of rat embryo cortical neurons at different times following preparation (D = days in vitro) to detect GAP43, SNAP25, PSD95, synaptophysin, and β-tubulin, as indicated; D4 + NL are cells incubated 24 h with 10 µg/mL NL between D3 and D4; D4, D5, and D6 are control cells incubated in the absence of NL and recovered on D4, D5 and D6. (**B**) Protein levels following densitometric analysis of immunoblots. Results are expressed as % of control (D4) (mean ± SD of 3 experiments using three different NL preparations on three different cell culture preparations. (* *p* < 0.05).

**Figure 4 marinedrugs-17-00406-f004:**
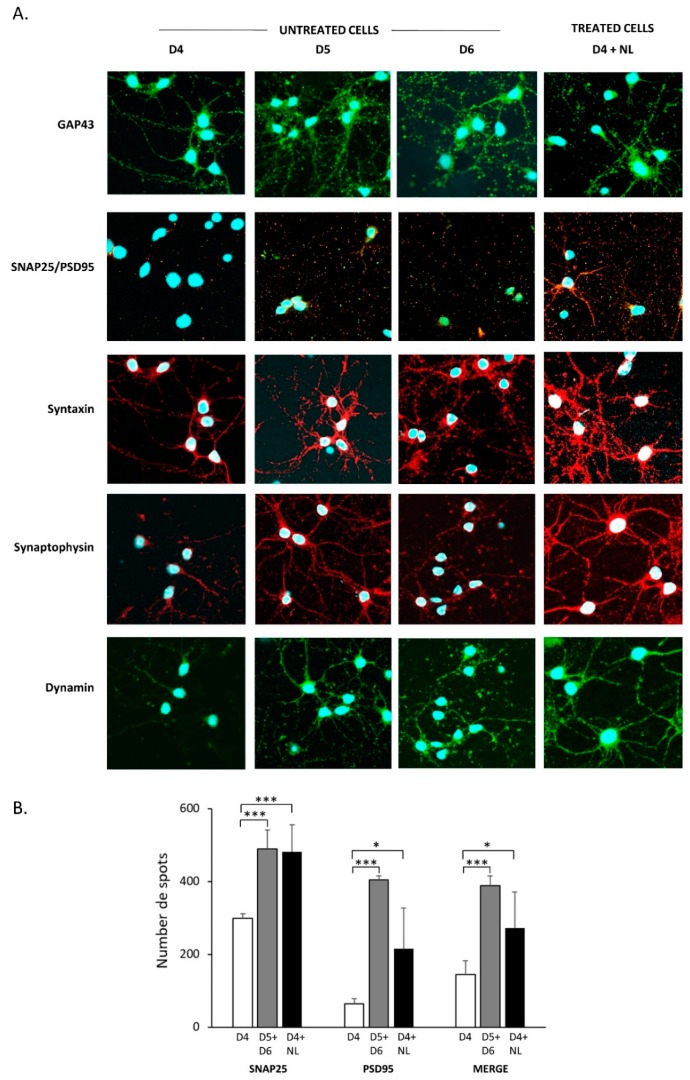
Cell localization of synaptic markers in primary cultures of cortical neurons treated with fish nanoliposomes (NL). (**A**). Immunofluorescence was used to detect different synaptic markers in primary cultures of rat embryo cortical neurons at the indicated times (D = days in vitro); D4 + NL are cells incubated 24 h with 10 µg/mL NL between D3 and D4; D4, D5, and D6 are control cells incubated in the absence of NL and recovered on D4, D5, and D6. Representative images are shown here obtained using the confocal microscope (120× magnification, Fluoview Fv10i, Olympus). (**B**). Image analysis of SNAP24/PSD95 labeling. The number of labeled spots of SNAP25 (red), PSD95 (green) and SNAP25 + PSD95 together (merge, orange), indicating colocalization are indicated for untreated cells on D4 (open bar, n = 4), on D5 and D6 (combined data shown as gray bar, n = 3), and for NL-treated cells on D4 (solid bar, n = 8). Statistically significant differences are shown as compared to the untreated cells (D4) for the corresponding label measured (* *p* < 0.05, *** *p* < 0.001).

**Table 1 marinedrugs-17-00406-t001:** Lipid classes and fraction of polar lipids composition of salmon lecithin.

Name	Salmon Lecithin
Total phospholipids (%)	67.7 ± 1.1
Phosphatidylcholine, PC (%)	42.4 ± 0.5
Phosphatidylethanolamine, PE (%)	7.7 ± 0.1
Phosphatidylserine, PS (%)	9.1 ± 0.1
Phosphatidylinositol, PI (%)	13.0 ± 0.3
Sphingomyelin, SPM (%)	1.5 ± 0.1
Lysophosphosphatidylcholine, LPC (%)	2.8 ± 0.1
Other phospholipids (%)	23.5 ± 0.2
Triglycerides, TAGs (%)	31.2 ± 0.8
Cholesterol, CHOL (%)	1.2 ± 0.1
Free fatty acids, FFA (%)	ND

**Table 2 marinedrugs-17-00406-t002:** Fatty acid composition of salmon lecithin.

Saturated Fatty Acids (SFA)		Monounsaturated Fatty Acids (MUFA)		Polyunsaturated Fatty Acids (PUFA)	
Fatty acid	% (SD) *	Fatty acid	% (SD)	Fatty acid	% (SD)
C14:0	2.72 (0.04)	C14:1n9	0.33 (0.02)	C18:2n6	1.20 (0.04)
C15:0	1.13 (0.01)	C16:1n7	3.23 (0.07)	C18:3n3	0.35 (0.01)
C16:0	20.67 (0.15)	C18:1n9	13.87 (0.13)	C20:4n6	3.52 (0.02)
C17:0	0.88 (0.05)	C20:1n9	2.11 (0.06)	C20:5n3 ^1^	11.03 (0.13)
C18:0	6.10 (0.05)	C22:1n9	0.79 (0.08)	C22:5n3	3.93 (0.33)
C20:0	0.35 (0.03)			C22:6n3 ^2^	26.26 (0.09)
C22:0	1.55 (0.06)				
SFA	33.40	MUFA	20.32	PUFA	46.28

* Mean (SD) % of total fatty acid composition of nanoliposomes (mean and standard deviation of triplicate determinations). ^1^ EPA, ^2^ DHA.

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
