# Peer review of "Neurotrophic Effect of Fish-Lecithin Based Nanoliposomes on Cortical Neurons"

_marinedrugs, 2019, doi:10.3390/md17070406_

Reviewer 1 Report

In the manuscript titled "Neurotrophic effect of fish-lecithin based nanoliposomes on cortical neurons”, the authors demonstrate that nanoliposomes from lecithin extracted from fish head by-products added in the culture media accelerate the development of neural networks in primary cultures of rat cortical neurons. Further, they propose that fish-derived NL, by virtue of their n-3 PUFA profile and neurotrophic effects, represent a new innovative bioactive vector for developing preventive or curative treatments of neurodegenerative diseases. The topic is very interesting and whole manuscript is well written and ease to understand. The experimental design is appropriate and adequately described. The results are clearly presented and support the conclusions of the authors.

I think that the paper is suitable for the pubblication on Marine drugs in the present form.

Author Response

The authors thank the referee for his positive feedback.

Reviewer 2 Report

I think it is an article with an important and interesting subject. However, it is a bit difficult to follow. The authors should better emphasize the importance of their work and the possible future applications and make the manuscript more ‘reader-friendly’. I recommend the publication of this article after corrections.

Introduction: the purpose of this work is not well explained. It should be highlighted (maybe add a paragraph explaining the objectives of your work). Please, provide a better explanation why did you choose nanoliposomes?

Other corrections:

Line 21: DHA- define abbreviation

Line 26: advanced stage of growth profile?? Did you mean the growth of axons? In what way it was similar to untreated cells? Please, rephrase, this sentence is not clear

Line 52: Provide references

Line 62: Did you develop this technique in this work, or has it been published before (in this case you should provide references, or if it hasn’t been published just mention unpublished data).

Line 67: provide references

Line 67: nanoliposomes offer (not offers)

Line 70: please, indicate cells types (2 examples)

Line 192: Please either provide more details about the extraction or reorganize section 3: in line 193 you mention about the solvent-free process (did you mean organic solvent-free?), and then you mention different solvents (probably used for a different purpose, but it is confusing). Also, you say that salmon lecithin was obtained by enzymatic hydrolysis…..The hydrolysis of which materials? Or maybe the salmon lecithin was the substrate, not the product? How was this material obtained/where purchased from? Please provide these details. On the other hand, you don’t need to mention that it was done in your laboratory. Does the cited reference 14 also cover the extraction process? How were the ‘lipidic fractions’ obtained? Was it that the lipidic fractions were obtained by the hydrolysis of salmon lecithin? Please, correct lines 192-193

Line 199: were the FAMEs isolated from lecithin or synthesized de novo?

Line 216: What mass/volume of ethanol?

Line 217: rotavapor- please, provide manufacturer/country

Line 219: sonicator type (probe/bath?), manufacturer/country

Line 220: the liposomes are not a solution! Please, use another word (dispersion/suspension?)

Lines 223: the electrophoretic mobility is not measured by DLS, but laser Doppler velocimetry (there are also other synonyms for this method). Although the instrument is the same the methods are different!

Line 226: did you mean the distribution of particle size?

Line 226: Did you mean: the electrophoretic mobility of dispersed particles?

Line 228: why did you choose these RI and absorbance values?

Line 229: please, rephrase this sentence

Line 236: described where? Do you mean section 3.4?

Line 242: how was the water removed from the sample? Were the samples analyzed immediately after 5 min incubation or maybe they were dried before?

Line 80: Table 1 caption: constituting of??? Please, change it. Maybe the better caption would be’ Composition of salmon lecithin’

Line 93: the viscosity of which material? How was the viscosity measured? Please, provide the viscosity values and how they affected the size (increased with increasing viscosity?) You must be careful with this statement, because the medium viscosity should be considered in the size measurements (in your case you diluted sample 200x with water, so the viscosity of dispersant should be the same as that of water, but if it is different the result should be corrected!)

Line 94-95: please provide more details about the effect of fatty acid composition, lipid profile and surface-active properties of lecithin on size

Line 97: what do you mean by ‘relatively high stability’. Please, be more precise

Line 98: Did you mean ‘the phosphate residues of lecithin molecules’?

Line 99: According to laser Doppler velocimetry results…. (DLS refers only to size/size distribution measurements)

Line 99: the electrophoretic mobility of what??? Of liposomes? Did you measure the electrophoretic mobility in water or in salmon lecithin? Please, correct this sentence

Lune 102-103: This reviewer does not fully agree with this statement. Zwitterionic phospholipids (similarly to other zwitterionic molecules) have a property referred to as the isoelectric point. The isoelectric point of PE is 4.20 for instance, and that of lecithin is 4.15 (Petelska and Figaszewski, Interfacial tension of bilayer lipid membrane formed from phosphatidylethanolamine. Biochim. Biophys. Acta, 1567 (2002), pp. 79-86), hence at physiological pH the negative charge should prevail despite the presence of both, phosphate and amino groups.

Line 113: In your schema, you should include the isolation of lecithin from head fish by-product, then then the formulation of nanoliposomes. The isolation of lecithin from head fish by-product and the information where this by-product was obtained from should be added to ‘materials and methods’ section.

Line 107-108: the bilayer nature of the vesicles is not visible in the TE micrographs! You would need cryo-TEM to observe this? Where did you observe the oil droplets and how did you determine the value 10%? If it is in TEM micrographs, how can you prove it’s oil and not staining effect?

Line 118-119: Please correct, it is not clear whether you mean that the release of active molecules that reflects the order and dynamics of phospholipid alkyl chains, or if it is the membrane fluidity?

Line 121-122: how does it keep the level of hydration?

Line 125: the permeant diffusion rate in what? Partitioning between what and what?

Line 131-132: what type of changes?

Line 133: what assay was used for cytotoxicity? MTT? LDH?

Line 134: did you mean increased expression after incubation with liposomes? Increased expression compared with what (what was the control?) Statistics? (p-values?)

Line 143: Which data? If it refers to data from line 134 onwards, then you can’t talk about increased expression. If there is no statistically significant difference, it means that the expression of markers was not affected. You should modify the text

Line 156: what was the control?

Lines 300-305- move to results and discussion?

Lines 310-312: The unpublished results cannot be considered as a conclusion from this work. Move to results and discussion. Do you have evidence of their improved fusogenic properties, or it is just a hypothesis based on the fact that they are liposomes?

Line 312: do you have the proof that they are multilayer vesicles? If not, delete this word

What do you mean by ‘using PUFA directly’- please, be more precise. Dissolving them in water???? Or is it that the liposomes provide a means of obtaining an aqueous dispersion of PUFAs with the particle size in the colloidal range, that otherwise would be difficult to obtain due to the low solubility of PUFAs in water? How the PUFAs or NLs could be delivered to their site of action (oral administration, parenteral administration???)- this should be included in the discussion.

I think the conclusions section should be shortened and should contain only the conclusions from the current study with a short indication of the future perspectives, and the rest should be moved to the results/discussion section.

Author Response

Line 21: DHA- define abbreviation. The abbreviation has been defined.

Line 26: advanced stage of growth profile?? Did you mean the growth of axons? In what way it was similar to untreated cells? Please, rephrase, this sentence is not clear The sentence has been modified to “Results revealed that NL-treated cells on day 4 displayed a level of neurite outgrowth and arborizationsimilar to those of untreated cells on day 5 and 6, 

Line 52: Provide references  References have been included as indicated below

It is clear that n-3 PUFA provide beneficial effects in neuronal development and growth by increasing membrane fluidity(Innis SM.. Dietary (n-3) fatty acids and brain development. J Nutr. 2007 137:855-859; Shindou H, Koso H, Sasaki J, Nakanishi H, Sagara H, Nakagawa KM, Takahashi Y, Hishikawa D, Iizuka-Hishikawa Y, Tokumasu F, Noguchi H, Watanabe S, Sasaki T, Shimizu T.Docosahexaenoic acid preserves visual function by maintaining correct disc morphology in retinal photoreceptor cells. J Biol Chem. 2017 292):12054-12064; Li D, Wahlqvist ML, Sinclair AJ. Advances in n-3 polyunsaturated fatty acid nutrition. Asia Pac J Clin Nutr. 2019 28:1-5.or as activators of cell signaling functions (Hishikawa D, Valentine WJ, Iizuka-Hishikawa Y, Shindou H, Shimizu T.Metabolism and functions of docosahexaenoic acid-containing membrane glycerophospholipids. FEBS Lett. 2017 591:2730-2744; Kitajka K, Sinclair AJ, Weisinger RS, Weisinger HS, Mathai M, Jayasooriya AP, Halver JE, Puskás LG. Effects of dietary omega-3 polyunsaturated fatty acids on brain gene expression)

Line 62: Did you develop this technique in this work, or has it been published before (in this case you should provide references, or if it hasn’t been published just mention unpublished data).

We used this technique for the previous articles. We added the number of patent in the text.

Line 67: provide references 

Maherani, B., Arab-Tehrany, E., Mozafari, M.R., Gaiani, C., Linder, M. (2011). A liposomes: review of manufacturing techniques and targeting strategies. Current Nanoscience, 7, 436-452.

Line 67: nanoliposomes offer (not offers)

We modified in the text.

Line 70: please, indicate cells types (2 examples)

We added in the text.

Line 192: Please either provide more details about the extraction or reorganize section 3: in line 193 you mention about the solvent-free process (did you mean organic solvent-free?), and then you mention different solvents (probably used for a different purpose, but it is confusing). Also, you say that salmon lecithin was obtained by enzymatic hydrolysis…..The hydrolysis of which materials? Or maybe the salmon lecithin was the substrate, not the product? How was this material obtained/where purchased from? Please provide these details. On the other hand, you don’t need to mention that it was done in your laboratory. Does the cited reference 14 also cover the extraction process? How were the ‘lipidic fractions’ obtained? Was it that the lipidic fractions were obtained by the hydrolysis of salmon lecithin? Please, correct lines 192-193

We cannot detail the extraction method. We added the patent number for this extraction in the text. We used the enzymatic extraction for hydrolyses the salmon heads by-product of fish.

Line 199: were the FAMEs isolated from lecithin or synthesized de novo?

The fatty acids composition was determined by using the Ackman method.  They were isolated from lecithin.

Line 216: What mass/volume of ethanol?

We used 10ml of ethanol

Line 217: rotavapor- please, provide manufacturer/country

Laborita 4000 Heidolph UK.

Line 219: sonicator type (probe/bath?), manufacturer/country

Sonicator type and manufacturer were added to the text.

Line 220: the liposomes are not a solution! Please, use another word (dispersion/suspension?)

We modified in the text.

Lines 223: the electrophoretic mobility is not measured by DLS, but laser Doppler velocimetry (there are also other synonyms for this method). Although the instrument is the same the methods are different!

Line 226: did you mean the distribution of particle size?

Distribution of particle size reports information about the size and range of a set of particles representative of a material

Line 226: Did you mean: the electrophoretic mobility of dispersed particles?

The nanoliposomes electrophoretic mobility

Line 228: why did you choose these RI and absorbance values?

We measured the RI of liposome

Line 229: please, rephrase this sentence

Line 236: described where? Do you mean section 3.4? 

We don’t understand your question

Line 242: how was the water removed from the sample? Were the samples analyzed immediately after 5 min incubation or maybe they were dried before?

The water removed by evaporation, the samples were analyzed immediately after 5min.

Line 80: Table 1 caption: constituting of??? Please, change it. Maybe the better caption would be’ Composition of salmon lecithin’

We modified in the text.

Line 93: the viscosity of which material? How was the viscosity measured? Please, provide the viscosity values and how they affected the size (increased with increasing viscosity?) You must be careful with this statement, because the medium viscosity should be considered in the size measurements (in your case you diluted sample 200x with water, so the viscosity of dispersant should be the same as that of water, but if it is different the result should be corrected!)

The viscosity of nanoliposome. The viscosity was measured by rheometer. In the previous works of our team, we measured the viscosity (Hasan, M., Ben Messaoud, G., Michaux, F., Tamayol, A., Kahn, c., Belhaj, N., Linder, M., Arab-Tehrany, E. (2016). Chitosan-coated liposomes encapsulating curcumin: Study lipid-polysaccharide interactions and investigate the rheological properties of the systems, RSC advances, 6, 45290-45304).

Line 94-95: please provide more details about the effect of fatty acid composition, lipid profile and surface-active properties of lecithin on size

We developed in the text.

Line 97: what do you mean by ‘relatively high stability’. Please, be more precise

With this value of electrophoretic mobility the stability of nanoliposome is higher than 30 days.

Line 98: Did you mean ‘the phosphate residues of lecithin molecules’?

Structure of polar lipid contains the phosphate element .

Line 99: According to laser Doppler velocimetry results…. (DLS refers only to size/size distribution measurements)

DLS were replaced in the text by electrophoretic light scattering (ELS).

Line 99: the electrophoretic mobility of what??? Of liposomes? Did you measure the electrophoretic mobility in water or in salmon lecithin? Please, correct this sentence

We modified in the text.

Lune 102-103: This reviewer does not fully agree with this statement. Zwitterionic phospholipids (similarly to other zwitterionic molecules) have a property referred to as the isoelectric point. The isoelectric point of PE is 4.20 for instance, and that of lecithin is 4.15 (Petelska and Figaszewski, Interfacial tension of bilayer lipid membrane formed from phosphatidylethanolamine. Biochim. Biophys. Acta, 1567 (2002), pp. 79-86), hence at physiological pH the negative charge should prevail despite the presence of both, phosphate and amino groups.

Line 113: In your schema, you should include the isolation of lecithin from head fish by-product, then then the formulation of nanoliposomes. The isolation of lecithin from head fish by-product and the information where this by-product was obtained from should be added to ‘materials and methods’ section.

The extraction of lecithin from salmon head was realized by using an enzymatic process, we added the patent number in the text and we cannot explain on detail.

Line 107-108: the bilayer nature of the vesicles is not visible in the TE micrographs! You would need cryo-TEM to observe this? Where did you observe the oil droplets and how did you determine the value 10%? If it is in TEM micrographs, how can you prove it’s oil and not staining effect?

By TEM we can observe the presence of nanoliposome. According the previous publication of our team,we observe the nanoliposomes and 10% of emulsion (Arab-Tehrany, E., Kahn, C.J.F., Baravian, C., Maherani, B., Belhaj, N., Wang, X., Linder, M. (2012). Elaboration and Characterization of Nanoliposome Made of Soya, Rapeseed and Salmon Lecithins: Application to Cell Culture. Colloids and Surfaces B: Biointerfaces. 95,75–81.

Line 118-119: Please correct, it is not clear whether you mean that the release of active molecules that reflects the order and dynamics of phospholipid alkyl chains, or if it is the membrane fluidity?

We added in the text. 

Line 121-122: how does it keep the level of hydration?

By increasing the percentage of unsaturated FAs, the packing between phospholipids decreases and keeps the level of hydration, thus maintaining membrane fluidity the presence of polar head will be allow the increasing the water trap by polar head and increasing the hydration.

 Line 125: the permeant diffusion rate in what? Partitioning between what and what?

Increase the permeant diffusion rate and partitioning tendency of active molecule and environment.

Line 131-132: what type of changes? We have clarified this point in the text: Since significant changes in the maturation process of the neuronal network are usually observed starting from D4 after plating,one group of cells were incubated with 10 µg/mL NL on D3. This concentration was selected based on absence of cytotoxicity (Figure 2). 

Line 133: what assay was used for cytotoxicity? MTT? LDH? Cytotoxicity was assessed using the MTT assay.  The cytotoxicity data has been included (Figure 2, see also response to referee 3)

Line 134: did you mean increased expression after incubation with liposomes? Increased expression compared with what (what was the control?) Statistics? (p-values?)Here, we are first discussing changes in protein levels over time (D5 and D6 as compared to D4) of different synaptic markers in control cells in the absence of NL. We have modified the text to clarify this point. 

Statistically significant differences and p values are indicated in the figure. Statistical methodology used (Student’s t-test) has been added to the methods section. 

Line 143: Which data? If it refers to data from line 134 onwards, then you can’t talk about increased expression. If there is no statistically significant difference, it means that the expression of markers was not affected. You should modify the text. We have modified the text and deleted any reference to increased expression.

Line 156: what was the control? We have modified the phrase, to indicate comparison of control cells of D5 and D6 relative to control cells at D4, and comparison of NL-treated cells at D4 relative to untreated control cells. 

Lines 300-305- move to results and discussion?  The authors thank the reviewer for this suggestion, and have moved this part to the Results and Discussion. 

Lines 310-312: The unpublished results cannot be considered as a conclusion from this work. Move to results and discussion. This section has been moved to the Results and Discussion (lines XXX). Do you have evidence of their improved fusogenic properties, or it is just a hypothesis based on the fact that they are liposomes? 

It is a hypothesis, we will validate by other analyses.

Line 312: do you have the proof that they are multilayer vesicles? If not, delete this word

By TEM we can observe the presence of nanoliposome. According the previous publication of our team,we observe the nanoliposomes and 10% of emulsion (Arab-Tehrany, E., Kahn, C.J.F., Baravian, C., Maherani, B., Belhaj, N., Wang, X., Linder, M. (2012). Elaboration and Characterization of Nanoliposome Made of Soya, Rapeseed and Salmon Lecithins: Application to Cell Culture. Colloids and Surfaces B: Biointerfaces. 95,75–81.

In addition, when we use the sonication technique, the nanoliposomes are multilayer.

What do you mean by ‘using PUFA directly’- please, be more precise. Dissolving them in water???? Or is it that the liposomes provide a means of obtaining an aqueous dispersion of PUFAs with the particle size in the colloidal range, that otherwise would be difficult to obtain due to the low solubility of PUFAs in water? How the PUFAs or NLs could be delivered to their site of action (oral administration, parenteral administration???)- this should be included in the discussion.

In this study, we used the mixture of various fatty acids (saturated, mono and poly unsaturated fatty acids) by using the natural lecithin from salmon head. In the literature, we observed that they use the pure phospholipids like as DOPC, DPPC and etc.

I think the conclusions section should be shortened and should contain only the conclusions from the current study with a short indication of the future perspectives, and the rest should be moved to the results/discussion section.

Reviewer 3 Report

This is an interesting study where the authors extracted fish lecithin and formed and characterized nanoliposomes (NL) and determined the effects of NL on cortical neurons. 

Some major concerns include:

1) The control treatment of neurons is not defined. Table 1 and 2 accurately define the NL composition. A great control would be using the same lecithin composition in Table 1 but adjusting the amounts of PUFA to a lower level. Treating neurons with the defined PUFA composition of the NL compared with treatment with NL with more saturated fatty acids will give better comparisons. Alternatively, the omega-3/8 ratios can be altered to 50 %, 25 %, and 0 % to test a specific hypothesis whether the effects on neural growth is mediated by the omega-3/6 ratio. As it stands, one is not sure what hypothesis was tested in these studies.

2) It is not clear why membrane fluidity (section 2.4) was determined for NL?  Is the fluidity related to the delivery to neurons?

3) On line 133, the authors indicate that 10 ng/mL LP was chosen due to cytotoxicity data. Please report the cytotoxicity data.

4) Could the lack of significance in neural protein expression be related to the fact that the control lipid composition was not defined? Also, control cells and NL treated cells were treated under different time scales. It would be preferable to have the same time points with defined lipid compositions that are being compared.

5) The authors indicate (lines 174-180) that image analyses “revealed a higher number” or “decreased slightly” or “increased numbers” but do not provide any quantification. Please provide quantification and statistical analyses..  

6) On lines 192-193, please provide details of the enzymatic hydrolysis and the low temperature process of obtaining salmon-lecithin.

Author Response

1) The control treatment of neurons is not defined. Table 1 and 2 accurately define the NL composition. A great control would be using the same lecithin composition in Table 1 but adjusting the amounts of PUFA to a lower level. Treating neurons with the defined PUFA composition of the NL compared with treatment with NL with more saturated fatty acids will give better comparisons. Alternatively, the omega-3/8 ratios can be altered to 50 %, 25 %, and 0 % to test a specific hypothesis whether the effects on neural growth is mediated by the omega-3/6 ratio. As it stands, one is not sure what hypothesis was tested in these studies.

The control condition is D4 in absence of NL, this has been clarified in the text and in the figure legends.  

Previous studies have tested the effects of individual fatty acids, with particular interest on the effect of n- fatty acids including DHA, shown to demonstrate potential neuroprotective effects in both cell and animal models. Here, our primary objective was to investigate the effects of fish-derived nanoliposomes on neurons, based on their high n-3 fatty acid content.  

2) It is not clear why membrane fluidity (section 2.4) was determined for NL?  Is the fluidity related to the delivery to neurons?

We specified in the text that it is a membrane fluidity of nanoliposome.

3) On line 133, the authors indicate that 10 ng/mL LP was chosen due to cytotoxicity data. Please report the cytotoxicity data.The cytotoxicity data has been included as an additional figure (Figure 2), and the text has been modified accordingly.  

4) Could the lack of significance in neural protein expression be related to the fact that the control lipid composition was not defined? Also, control cells and NL treated cells were treated under different time scales. It would be preferable to have the same time points with defined lipid compositions that are being compared.

Here, we show the results obtained with three different cultures of cortical neurons, and three different lots of NL, which could contribute to the high inter-variability, and thus to the fact that statistical significance was not achieved. Western blots are only semi-quantitative, and signal intensity can vary considerably between blots, which could also contribute to this issue.  Nevertheless, for each experiment, we have consistently observed the similar trend where the NL-treated cells at D4 demonstrate a profile of protein markers and morphology comparable to that of control cells at D5 and D6, as demonstrated by the representative immunoblots shown in Figure 3. 

Control and NL-treated cells were treated in an identical manner with respect to the time frame. Control cells were evaluated on D4, D5 and D6.  NL-treated cells were analyzed on D4 after 24h incubation with NL (between D3 and D4). 

5) The authors indicate (lines 174-180) that image analyses “revealed a higher number” or “decreased slightly” or “increased numbers” but do not provide any quantification. Please provide quantification and statistical analyses.  

We have added these results as a bar graph in Figure 4, and also modified the text.      

6) On lines 192-193, please provide details of the enzymatic hydrolysis and the low temperature process of obtaining salmon-lecithin.

We cannot explain in the article the different steps of this process. We have a patent (patent n° FR 2.835.703) for this enzymatic process.

Round  2

Reviewer 3 Report

Please see the comments highlighted in yellow. Thanks

1) The control treatment of neurons is not defined. Table 1 and 2 accurately define the NL composition. A great control would be using the same lecithin composition in Table 1 but adjusting the amounts of PUFA to a lower level. Treating neurons with the defined PUFA composition of the NL compared with treatment with NL with more saturated fatty acids will give better comparisons. Alternatively, the omega-3/8 ratios can be altered to 50 %, 25 %, and 0 % to test a specific hypothesis whether the effects on neural growth is mediated by the omega-3/6 ratio. As it stands, one is not sure what hypothesis was tested in these studies.

The control condition is D4 in absence of NL, this has been clarified in the text and in the figure legends.  

Previous studies have tested the effects of individual fatty acids, with particular interest on the effect of n- fatty acids including DHA, shown to demonstrate potential neuroprotective effects in both cell and animal models. Here, our primary objective was to investigate the effects of fish-derived nanoliposomes on neurons, based on their high n-3 fatty acid content. 

Please cite these studies and indicate that these are relevant controls to the present study? Did the other authors use the same systems with the same endpoints?

2) It is not clear why membrane fluidity (section 2.4) was determined for NL?  Is the fluidity related to the delivery to neurons?

We specified in the text that it is a membrane fluidity of nanoliposome.

Could you indicate the significance of the membrane fluidity on the study? Does a change in fluidity affect delivery of the NL?. 

3) On line 133, the authors indicate that 10 ng/mL LP was chosen due to cytotoxicity data. Please report the cytotoxicity data.The cytotoxicity data has been included as an additional figure (Figure 2), and the text has been modified accordingly. 

Thanks

4) Could the lack of significance in neural protein expression be related to the fact that the control lipid composition was not defined? Also, control cells and NL treated cells were treated under different time scales. It would be preferable to have the same time points with defined lipid compositions that are being compared.

Here, we show the results obtained with three different cultures of cortical neurons, and three different lots of NL, which could contribute to the high inter-variability, and thus to the fact that statistical significance was not achieved. Western blots are only semi-quantitative, and signal intensity can vary considerably between blots, which could also contribute to this issue.  Nevertheless, for each experiment, we have consistently observed the similar trend where the NL-treated cells at D4 demonstrate a profile of protein markers and morphology comparable to that of control cells at D5 and D6, as demonstrated by the representative immunoblots shown in Figure 3. 

Control and NL-treated cells were treated in an identical manner with respect to the time frame. Control cells were evaluated on D4, D5 and D6.  NL-treated cells were analyzed on D4 after 24h incubation with NL (between D3 and D4).

This is the weak part of this study and this raises a concern that appropriate controls were not used. Comparing studies at different time points under different incubation conditions will be expected to yield variable data. For each of the three studies, could the data be normalized to baseline?

5) The authors indicate (lines 174-180) that image analyses “revealed a higher number” or “decreased slightly” or “increased numbers” but do not provide any quantification. Please provide quantification and statistical analyses.  

We have added these results as a bar graph in Figure 4, and also modified the text.  

Thank you.

6) On lines 192-193, please provide details of the enzymatic hydrolysis and the low temperature process of obtaining salmon-lecithin.

We cannot explain in the article the different steps of this process. We have a patent (patent n° FR 2.835.703) for this enzymatic process.

For peer review, it is difficult to access the validity of a method or process that is not adequately described.

Author Response

1) The control treatment of neurons is not defined. Table 1 and 2 accurately define the NL composition. A great control would be using the same lecithin composition in Table 1 but adjusting the amounts of PUFA to a lower level. Treating neurons with the defined PUFA composition of the NL compared with treatment with NL with more saturated fatty acids will give better comparisons. Alternatively, the omega-3/8 ratios can be altered to 50 %, 25 %, and 0 % to test a specific hypothesis whether the effects on neural growth is mediated by the omega-3/6 ratio. As it stands, one is not sure what hypothesis was tested in these studies.

The control condition is D4 in absence of NL, this has been clarified in the text and in the figure legends.  

Previous studies have tested the effects of individual fatty acids, with particular interest on the effect of n- fatty acids including DHA, shown to demonstrate potential neuroprotective effects in both cell and animal models. Here, our primary objective was to investigate the effects of fish-derived nanoliposomes on neurons, based on their high n-3 fatty acid content. 

Please cite these studies and indicate that these are relevant controls to the present study? Did the other authors use the same systems with the same endpoints?

Two studies were cited (5,6) in the introduction. The markers here are consistent with results obtained using DHA or other fatty acids in these previous studies who used endpoints including cell viability tests, immunohistology (GAP43), and markers of differentiation. 

Sakamoto T, Cansev M, Wurtman RJ. Oral supplementation with docosahexaenoic acid and uridine-5'-monophosphate increases dendritic spine density in adult gerbil hippocampus. Brain Res. 1182:50-59, 2007. 

Kawakita E, Hashimoto M, Shido O. Docosahexaenoic acid promotes neurogenesis in vitro and in vivo. Neuroscience. 139:991-997, 2006.

These have been added to the text as well

2) It is not clear why membrane fluidity (section 2.4) was determined for NL?  Is the fluidity related to the delivery to neurons?

We specified in the text that it is a membrane fluidity of nanoliposome.

Could you indicate the significance of the membrane fluidity on the study? Does a change in fluidity affect delivery of the NL?.  

The transfer of active molecule depends on various parameters like as fluidity of lipidic membrane of nanoliposomes. The presence of PUFA will be accelerate the transfer phenomenon.

3) On line 133, the authors indicate that 10 ng/mL LP was chosen due to cytotoxicity data. Please report the cytotoxicity data.The cytotoxicity data has been included as an additional figure (Figure 2), and the text has been modified accordingly. 

Thanks

4) Could the lack of significance in neural protein expression be related to the fact that the control lipid composition was not defined? Also, control cells and NL treated cells were treated under different time scales. It would be preferable to have the same time points with defined lipid compositions that are being compared.

Here, we show the results obtained with three different cultures of cortical neurons, and three different lots of NL, which could contribute to the high inter-variability, and thus to the fact that statistical significance was not achieved. Western blots are only semi-quantitative, and signal intensity can vary considerably between blots, which could also contribute to this issue.  Nevertheless, for each experiment, we have consistently observed the similar trend where the NL-treated cells at D4 demonstrate a profile of protein markers and morphology comparable to that of control cells at D5 and D6, as demonstrated by the representative immunoblots shown in Figure 3. 

Control and NL-treated cells were treated in an identical manner with respect to the time frame. Control cells were evaluated on D4, D5 and D6.  NL-treated cells were analyzed on D4 after 24h incubation with NL (between D3 and D4).

This is the weak part of this study and this raises a concern that appropriate controls were not used. Comparing studies at different time points under different incubation conditions will be expected to yield variable data. For each of the three studies, could the data be normalized to baseline?

The authors reaffirm that for each set of cells incubated with NL, a corresponding set of cells were incubated in the absence of nanoliposomes (D4).  For each experiments, data was normalized for each corresponding D4 control. Estimated protein levels based on densitrometric analysis for D4 are indicated at 100% for immunoblots. Statistical analysis was performed only after normalization We have indicated this in the text (Materials and Methods, statistical analysis) to avoid any confusion. 

5) The authors indicate (lines 174-180) that image analyses “revealed a higher number” or “decreased slightly” or “increased numbers” but do not provide any quantification. Please provide quantification and statistical analyses.  

We have added these results as a bar graph in Figure 4, and also modified the text.  

Thank you.

6) On lines 192-193, please provide details of the enzymatic hydrolysis and the low temperature process of obtaining salmon-lecithin.

We cannot explain in the article the different steps of this process. We have a patent (patent n° FR 2.835.703) for this enzymatic process.

For peer review, it is difficult to access the validity of a method or process that is not adequately described.

I am sorry but we cannot detail this part.